# Enantioselective synthesis of chiral α,α-dialkyl indoles and related azoles by cobalt-catalyzed hydroalkylation and regioselectivity switch

Jiangtao Ren[1,2,5], Zheng Sun[1,5], Shuang Zhao[1,3], Jinyuan Huang[1,3], Yukun Wang[1], Cheng Zhang[1,3], Jinhai Huang[1], Chenhao Zhang[1], Ruipu Zhang[1,3], Zhihan Zhang[4] ✉, Xu Ji[1,3] ✉ & Zhihui Shao [1,2] ✉

General, catalytic and enantioselective construction of chiral α,α-dialkyl indoles represents an important yet challenging objective to be developed. Herein we describe a cobalt catalyzed enantioselective *anti*-Markovnikov alkene hydroalkylation via the remote stereocontrol for the synthesis of α,α-dialkyl indoles and other *N*-heterocycles. This asymmetric C(sp³)−C(sp³) coupling features high flexibility in introducing a diverse set of alkyl groups at the α-position of chiral *N*-heterocycles. The utility of this methodology has been demonstrated by late-stage functionalization of drug molecules, asymmetric synthesis of bioactive molecules, natural products and functional materials, and identification of a class of molecules exhibiting anti-apoptosis activities in UVB-irradiated HaCaT cells. Ligands play a vital role in controlling the reaction regioselectivity. Changing the ligand from bi-dentate **L6** to tridentate **L12** enables CoH-catalyzed Markovnikov hydroalkylation. Mechanistic studies disclose that the *anti*-Markovnikov hydroalkylation involves a migratory insertion process while the Markovnikov hydroalkylation involves a MHAT process.

The development of catalytic and enantioselective methods to construct a stereogenic center bearing two different alkyl groups is of great interest owing to their ubiquitous occurrence in a wide array of natural products and pharmacologically active molecules as well as their wide applications as versatile intermediates in synthesis. Among them, chiral α,α-dialkyl indoles **I**, in which a stereogenic carbon-bearing two alkyl substituents are bound to the indole *N*, are subunits in pharmaceuticals and bioactive molecules, chiral ligands as well as functional materials (Fig. 1a)[1–7]. Notably, GSK126 (**A**) is a small molecule inhibitor of EZH2 methyltransferase activity with a chiral α,α-dialkyl indole skeleton, which is highly selective and efficient. It can effectively reduce global H3K27me3 levels and reactivates silenced PRC2 target genes[1]. Compound **B** has the potential for the treatment of uterine fibroids based on its ability to act as a potent progesterone receptor (PR) antagonist[2]. Compound **D** acted as an efficient chiral ligand in catalytic asymmetric allylations and hydrogenations[5]. These molecules were synthesized by chiral resolution or utilizing stoichiometric chiral compounds. The development of general, catalytic and

[1]Key Laboratory of Medicinal Chemistry for Natural Resource, Ministry of Education, School of Chemical Science and Technology, State Key Laboratory for Conservation and Utilization of Bio-Resources in Yunnan, Yunnan University, 650500 Kunming, China. [2]Southwest United Graduate School, 650092 Kunming, China. [3]School of Pharmacy, Yunnan University, 650500 Kunming, China. [4]College of Chemistry, Central China Normal University, 430079 Wuhan, China. [5]These authors contributed equally: Jiangtao Ren, Zheng Sun. ✉e-mail: zhihanzhang@ccnu.edu.cn; jixu@ynu.edu.cn; zhihui_shao@hotmail.com

**Fig. 1 | Background and this work. a** Selected chiral α, α-dialkyl indole molecules. **b** Metal hydride (MH)-catalyzed enantioselective hydroalkylation of alkenes. **c** This work. CoH = cobalt hydride.

enantioselective methods to construct chiral α,α-dialkyl indoles **I** is highly desirable. Such a development would also facilitate to discover applications of such molecules.

The development of catalytic enantioselective methods to generate a α-stereogenic carbon center to the indole *N* is challenging, and

various reactions based on the direct C-N formation strategy using 1H-indoles[8–13] or the indirect strategy using *N*-substituted indoles or indole precursors[14–17] have been developed elegantly in the past few years. However, these processes mainly target indoles with a α-stereogenic carbon center bearing a sp2- or sp-hybridized carbon substituent.

**Table 1 | Comparison of Ni-H and Co-H catalyzed enantioselective *anti*-Markovnikov hydroalkylation of alkenes[a]**

| Entry | Metal salt | Solvent | Yield (%) | er |
|---|---|---|---|---|
| 1 | NiBr$_2$·DME | DME | 9 | 69:31 |
| 2 | NiBr$_2$·DME | THF | trace | N.D. |
| 3 | NiBr$_2$·DME | DMA | trace | N.D. |
| 4 | NiBr$_2$·DME | DMF | trace | N.D. |
| 5 | CoBr$_2$·DME | DME | 98 | 95:5 |
| 6 | CoBr$_2$·DME | THF | 75 | 95:5 |
| 7 | CoBr$_2$·DME | DMA | 48 | 57:43 |

[a]1a (0.10 mmol), 2a (0.20 mmol, 2.0 equiv), metal salt (10 mol%), L6 (15 mol%), CsF (0.20 mmol, 2.0 equiv), DEMS (0.20 mmol, 2.0 equiv), solvent (0.5 mL, 0.2 M), 0 °C, 24 h. Isolated yields. Er value was determined by chiral HPLC analysis. DEMS = diethoxymethylsilane.

Moreover, the reactions using 1H-indoles mainly rely on the use of indoles with electron-withdrawing groups or with substituents blocking the C3 position. Strategic bond-forming reactions to general, catalytic, and enantioselective assembly of chiral $\alpha,\alpha$-dialkyl indoles, including those with two minimally differentiated alkyl substituents, represents an important yet challenging objective to be developed.

Metal hydride (MH)-catalyzed enantioselective reductive hydroalkylation of alkenes with alkyl electrophiles has emerged as an attractive strategy toward value-added chemicals and medicinally relevant compounds, which allows for simultaneous construction of alkyl–alkyl (C(sp$^3$)–C(sp$^3$)) bond and generation of a new stereocenter[18–22]. Depending on the type of metal, it can be categorized as copper-hydride[23], nickel-hydride[24–42], cobalt-hydride[43–49], and iron-hydride[50]. The direct use of alkenes, which are stable and readily available, as pro-nucleophiles, obviates the need for preformed (over) stoichiometric amounts of moisture- and air-sensitive organometallic reagents, and offers significant advantages in practicality, scope, and functional group compatibility over traditional alkyl–alkyl cross-coupling[51]. However, to our knowledge, no such method has been described for the catalytic enantioselective synthesis of chiral indoles with a $\alpha$-stereogenic carbon center to the indole *N*. Given the importance of chiral $\alpha,\alpha$-dialkyl indoles and the ready availability of IV[52], metal hydride-catalyzed enantioselective reductive hydroalkylation of 1,1-disubstituted alkenes IV was envisioned to provide a general and efficient route to chiral $\alpha,\alpha$-dialkyl indoles (Fig. 1c, left). However, in contrast with the progresses made with mono-substituted and 1,2-disubstituted alkenes involving a branched (secondary) alkyl metal intermediates of the type II (Fig. 1b)[23–50], 1,1-disubstituted alkenes have rarely been successfully employed in metal-hydride-catalyzed enantioselective reductive hydroalkylation reactions. The key challenge is the need to identify a catalytic system that is capable of not only efficient and regio-selective M-H insertion of these sterically encumbered substrates but also effectively distinguish both substituents on alkenes far from the metal center and the chiral ligand. Thus, catalytic asymmetric functionalization of 1,1-disubstituted alkenes remains a long-standing challenge, and only a few examples of highly enantioselective hydrofunctionalization have been reported[53–55].

Herein we establish a chiral cobalt hydride catalytic system to achieve asymmetric hydroalkylation of 1,1-disubstituted alkenes IV to assemble a wide range of $\alpha,\alpha$-dialkyl indoles with a high level of *anti*-Markovnikov regioselectivity and enantioselectivity (Fig. 1c, left). The method features high flexibility in introducing a diverse set of alkyl groups at the $\alpha$-position of chiral indoles and is able to produce chiral indoles with two minimally differentiated $\alpha$-alkyl substituents, a task difficult to achieve using existing methods. Moreover, this CoH-catalyzed enantioselective hydro-alkylation can also be applied for the synthesis of other chiral $\alpha,\alpha$-dialkyl heterocycles such as valuable pyrroles and carbazoles. A modular, catalytic and enantioselective method to $\alpha,\alpha$-dialkyl heterocycles including indoles, pyrroles and carbazoles remains elusive. The method operates under mild conditions and shows broad functional group tolerance and uses readily accessible precursors. The utility of this methodology has been demonstrated by late-stage functionalization of drugs and catalytic enantioselective synthesis of EZH2 inhibitor GSK126, natural products (-)-Indolizidine 167B and (-)-Monomorine as well as conducting polymers. Notably, it also enable us to discover a class of molecules exhibiting anti-apoptosis activities in UVB-irradiated HaCaT cells.

Another attractive feature of the current methodology is that the use of Co-H catalysis enable the regioselectivity switch towards Markovnikov hydroalkylation of 1,1-disubstituted alkenes IV by changing the ligand from bidentate L6 to tridentate L12

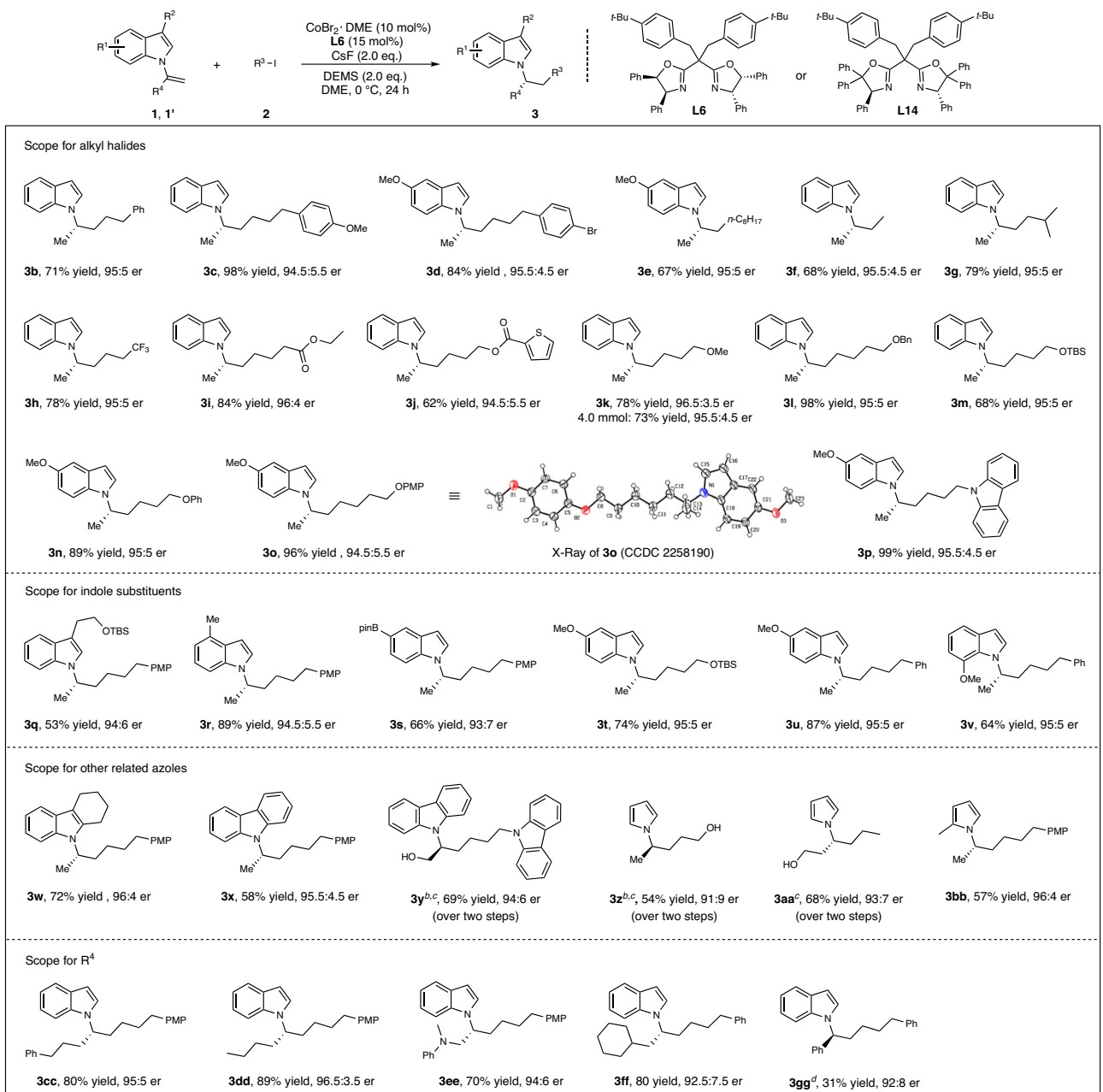

**Fig. 2 | CoH-catalyzed enantioselective *anti*-Markovnikov hydroalkylation to access chiral α,α-dialkyl indoles and related azoles.** Condition A: [a]All reactions were carried out with **1** or **1'** (0.10 mmol), **2** (0.20 mmol, 2.0 equiv), CoBr₂·DME (10 mol%), **L6** (15 mol%), CsF (0.20 mmol, 2.0 equiv), DEMS (0.20 mmol, 2.0 equiv),

DME (0.5 mL, 0.2 M), 0 °C, 24 h. Isolated yields. Er value was determined by chiral HPLC analysis. [b]*ent*-**L6** instead of **L6**. [c]TBAF (1.0 mL, 1.0 M in THF), THF (1.0 mL), rt, 2 h. [d]*ent*-**L14** instead of **L6**. PMP = 4-MeO-C₆H₄, TBS = *t*-butyldimethylsilyl.

(Fig. 1c, right). Indoles bearing a α-quaternary center were obtained with high regioselectivity. Notably, mechanistic studies disclose that the Markovnikov hydroalkylation of 1,1-disubstituted alkenes **IV** proceeded through a metal-hydride hydrogen atom transfer (MHAT) pathway, which is different from *anti*-Markovnikov hydroalkylation involving a Co-H migratory insertion process. Because of steric hindrance, facile β-hydrogen elimination, and ease of isomerization of tertiary alkylmetals, metal hydride-catalyzed sterically-disfavored Markovnikov hydroalkylation of 1,1-disubstituted alkenes remains an elusive challenge even in a racemic fashion[56]. Meanwhile, there are no strategies available to address the regioselectivity control to achieve the metal hydride catalyzed regiodivergent hydroalkylation of 1,1-disubstituted alkenes.

Cobalt as an abundant and low-toxicity transition metal, plays an important role in organic synthesis[57]. Hydroformylation, hydroboration and hydrosilylation of alkenes have been realized using low-valent cobalt-hydride species such as Co(I)H or Co(II)H[58–60]. However, Co(I)H or Co(II)H-catalyzed alkene hydroalkylation to form C(sp³)-C(sp³) bond has been under-exploited, and regiodivergent alkene hydroalkylation via low-valent Co(I)H or Co(II)H catalysis has not been achieved. In contrast with impressive advances with Ni-H catalysis, there are only a few reports of Co(I)H or Co(II)H-catalyzed enantioselective alkene hydroalkylation so far[44–49]. In these reports, 1,2-disubstituted alkenes were used as substrates, and branched (secondary) alkyl Co intermediates of type **II** were involved[44–49]. To the best of our knowledge, Co(I)H or Co(II)H-catalysis has not been employed in the hydroalkylation of 1,1-disubstituted alkenes. Co(I)H or Co(II)H-catalysis may

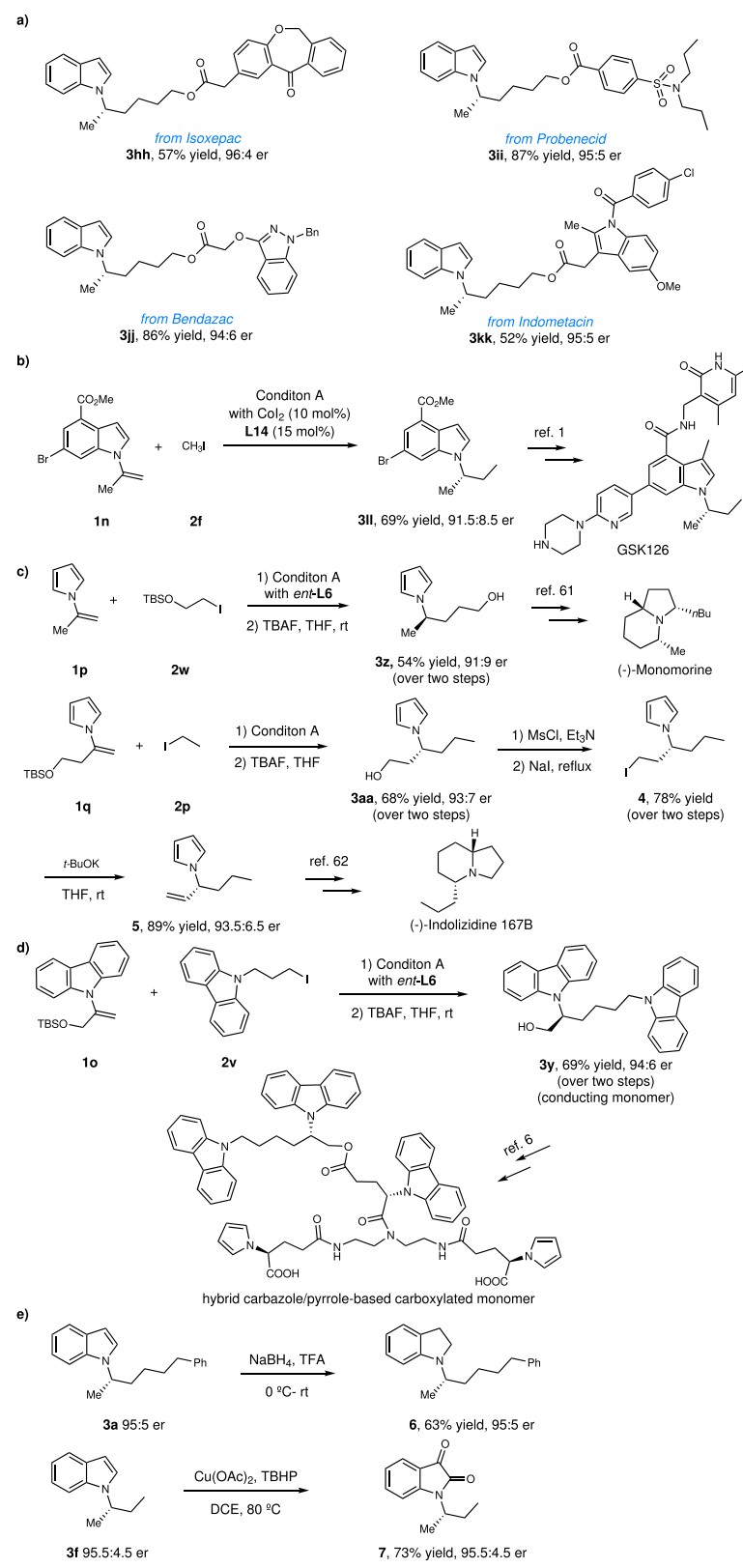

**Fig. 3 | Synthetic applications and transformations. a** Diversification of drug molecules. **b** Catalytic asymmetric synthesis of EZH2 inhibitor GSK126. **c** Catalytic asymmetric formal synthesis of natural prodcuts (-)-Monomorine and (-)-Indolizidine 167B. **d** Catalytic asymmetric synthesis of conducting monomers. **e** Synthetic transformations. TBAF = tetranbutylammoniumfluoride. MsCl = methanesulfonyl chloride. TBHP = tert-butyl hydroperoxide.

provide a complementary mode to Ni-H catalysis to address challenging substrate type/scope and regioselectivity and enantioselectivity control issues. Different from 1,2-disubstituted alkenes[44–49], Co(I)H or Co(II)H-catalyzed enantioselective *anti*-Markovnikov hydroalkylation of 1,1-disubstituted alkenes would involve linear alkyl Co intermediates of type **III**, and would require remote stereocontrol, whereas Markovnikov hydroalkylation process would involve sterically encumbered tertiary alkyl Co intermediates. A high-valent Co(IV)H-catalyzed

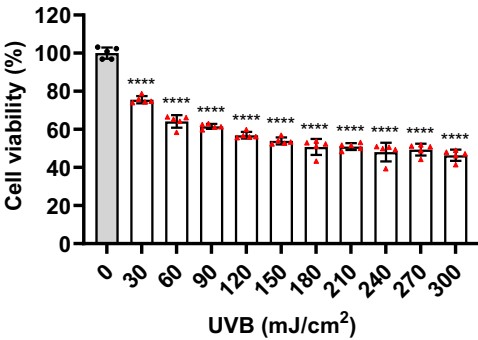

**Fig. 4 | UVB irradiation could injure HaCaT cells.** HaCaT cells were exposed to different doses of UVB (0–300 mJ/cm$^2$). After that, the cells were incubated with MTT for 4 h. Cell viability was measured by the absorbance of dissolved MTT crystals using an ELISA reader ($n = 5$). All data were analyzed by software GraphPad Prism 9.5.0 and presented as mean ± SD from three independent experiments. Statistical significance between the indicated groups were calculated by using one-way ANOVA and multiple comparisons. Adjustments were made for multiple comparisons. Significance is presented as ****$p < 0.0001$ (adjusted $p$ value) versus the control (0 mJ/cm$^2$) group.

anti-Markovnikov non-enantioselective hydroalkylation of alkenes with oxime esters was recently reported by introducing an 8-aminoquinoline as the directing group[43]. In this directed process, alkyl Co intermediates were not be involved.

## Results

### Reaction development

In initial studies, we applied nickel catalysts for the hydroalkylation reaction of **1a** and **2a**, given that Ni-H catalysis has found wide application in the alkene hydroalkylation. Unfortunately, we obtained only 9% yield and 69:31 er of the anti-Markovnikov hydroalkylation product **3a** with DME as the solvent (Table 1, entry 1). Only a trace of product **3a** was detected by using THF, DMA and DMF, which are frequently used solvents in Ni-H catalysis (entries 2-4). We therefore turned our attention to the use of other metal salts, and found that CoBr$_2$·DME as the metal salt and DME as the solvent in combination with the chiral ligand **L6** offered the anti-Markovnikov hydroalkylation product **3a** in 98% yield with 95:5 er (entry 5) (see Supplementary Table 1 for the details).

Under the optimized conditions, we investigated the substrate scope of this reaction (Fig. 2). Firstly, a wide range of alkyl iodides smoothly underwent the anti-Markovnikov hydroalkylation to produce the chiral $\alpha,\alpha$-dialkyl indoles (**3a-3p**) in 62-99% yield with 94.5:5.5-96.5:3.5 er. Scale-up synthesis was achieved on the scale of 4.0 mmol to give the desired product **3k** (675.4 mg) in 73% yield and 95.5:4.5 er. It is worth noting that iodomethane can also be used for this reaction, providing product **3f** with a stereocenter bearing two small yet similar methyl and ethyl group. Iodocyclohexane was found to be less reactive (see Supplementary Fig. 5). The configuration of the product **3o** was confirmed by X-ray crystallographic analysis. Next, we examined the scope of the indole substituents. A variety of indole substituents at 3-position (**3q**), 4-position (**3r**), 5-position (**3t-3u**) and 7-position (**3v**) were readily accommodated. Notably, a pinacol boronate moiety (Bpin, **3s**) can be tolerated, thus holding promising potential for medicinal chemistry applications. In addition, other N-heterocycles such as tetrahydrocarbazole (**3w**), carbazole (**3xx-3yy**) and pyrrole (**3z-3bb**) also underwent the hydroalkylation to offer the corresponding products with good er values. With respect to the alkene substituents (R$^4$), the hydroalkylation proceeded smoothly, delivering the desired indoles with $\alpha$-stereocenter bearing two diverse alkyl groups (**3cc-3gg**) in 31-89% yield with 92:8-96.5:3.5 er.

## Applications

The present method has wide applications. It can be applied in diversification of drug molecules (Fig. 3a, **3hh-3kk**). The present method can also be used for the catalytic enantioselective synthesis of $\alpha,\alpha$-dialkyl indole **3ll**, a key intermediate for the synthesis of EZH2 inhibitor GSK126 (Fig. 3b)[1]. In addition, the present method can also be used for the catalytic asymmetric synthesis of natural products. For examples, chiral $\alpha,\alpha$-dialkyl pyrrole **3z** obtained by our method is the key intermediate in the asymmetric synthesis of (-)-Monomorine[61], thus constituting a formal catalytic asymmetric synthesis of (-)-Monomorine (Fig. 3c). It is noted that **3z** was previously synthesized by stoichiometric chiral method[61]. Chiral $\alpha,\alpha$-dialkyl pyrrole **3aa** obtained by our method can be converted to compound **5** through halogenation and elimination. **5** is the key intermediate for the asymmetric synthesis of (-)-Indolizidine 167B[62]. Chiral $\alpha,\alpha$-dialkyl carbazole **3y** obtained by our method can be used as a conducting monomer and subsequently polymerized into a conducting polymer for electrochemical studies (Fig. 3d)[7]. It is noted that **3y** was previously synthesized by stoichiometric chiral method[7]. Meanwhile, hybrid carbazole/pyrrole-based carboxylated monomer can be synthesized from **3y**[6]. Chiral $\alpha,\alpha$-dialkyl indoles obtained by our method can be easily converted into other valuable heterocycles (Fig. 3e). For example, **3a** can be converted to chiral $\alpha,\alpha$-dialkyl indoline **6** by reduction with NaBH$_4$ (Fig. 3e). **3f** can be oxidized to chiral $\alpha,\alpha$-dialkyl isatin **7**.

The skin serves as a crucial barrier to safeguard the body against environmental stressors. Nonetheless, exposure to ultraviolet radiation (UV) and diverse environmental oxidative stressors may result in skin inflammation[63,64]. Indole derivatives, including melatonin and indole-3-lactic acid (ILA), have the potential to mitigate the cytotoxicity induced by UVB irradiation and suppress the generation of pro-inflammatory cytokines in UVB-irradiated keratinocytes[65–67]. In this study, we synthesized a series of N-alkylated indoles, but it remained unknown whether they have photoprotective effects on UVB-induced skin cell damage. To detect the protective effects of $\alpha,\alpha$-dialkyl indoles on UV induced skin injury, we firstly established a UVB-injured skin cell model. The HaCaT cells were irradiated by different doses of UVB (0-300 mJ/cm$^2$). Then the cell viability of the HaCaT cells was examined. The results showed that the cell viability of the HaCaT cells was gradually decreased with the increment of the UVB irradiation doses (Fig. 4), which suggesting that UV exposure could injure skin derived HaCaT cells. Since 150 mJ/cm$^2$ UVB irradiation could cause near 50% HaCaT cells damage (Fig. 4), we used this UVB dose in our subsequent experiments.

Next, we tested the effects of $\alpha,\alpha$-dialkyl indoles on UVB-induced damage in HaCaT cells. The HaCaT cells were treated with 27 $\alpha,\alpha$-dialkyl indoles or EGCG (positive control) at 35 μg/mL respectively for 6 hours after 150 mJ/cm$^2$ UVB irradiation. The results showed that 10 compounds, such as **3f, 3j, 3l, 3n, 3p, 3q, 3s, 3u, 3cc, 3ee**, significantly reverted UVB induced HaCaT cells injury, which were comparable or more better than the positive drugs EGCG (Fig. 5a). Additionally, we further examined the influences of these 10 compounds on the cell viability of HaCaT cells, and found that most of them (7 of 10) could apparently enhanced the cell viability of HaCaT cells (Fig. 5b). Altogether, these data indicated that $\alpha,\alpha$-dialkyl indoles probably have potential protective effects against UVB-induced damage in HaCaT human keratinocytes due to their obviously improvements on the cell viability.

### Regioselectivity switch

During the course of CoH-catalyzed hydroalkylation reaction, we found that tridentate ligand **L12** switched the reaction regioselectivity exclusively to lead to sterically-disfavored Markovnikov hydroalkylation product **8a** in 79% yield (Fig. 6). Interestingly, benzyl chlorides were able to provide the corresponding Markovnikov hydroalkylation products **8r-8u** in acceptable yields,

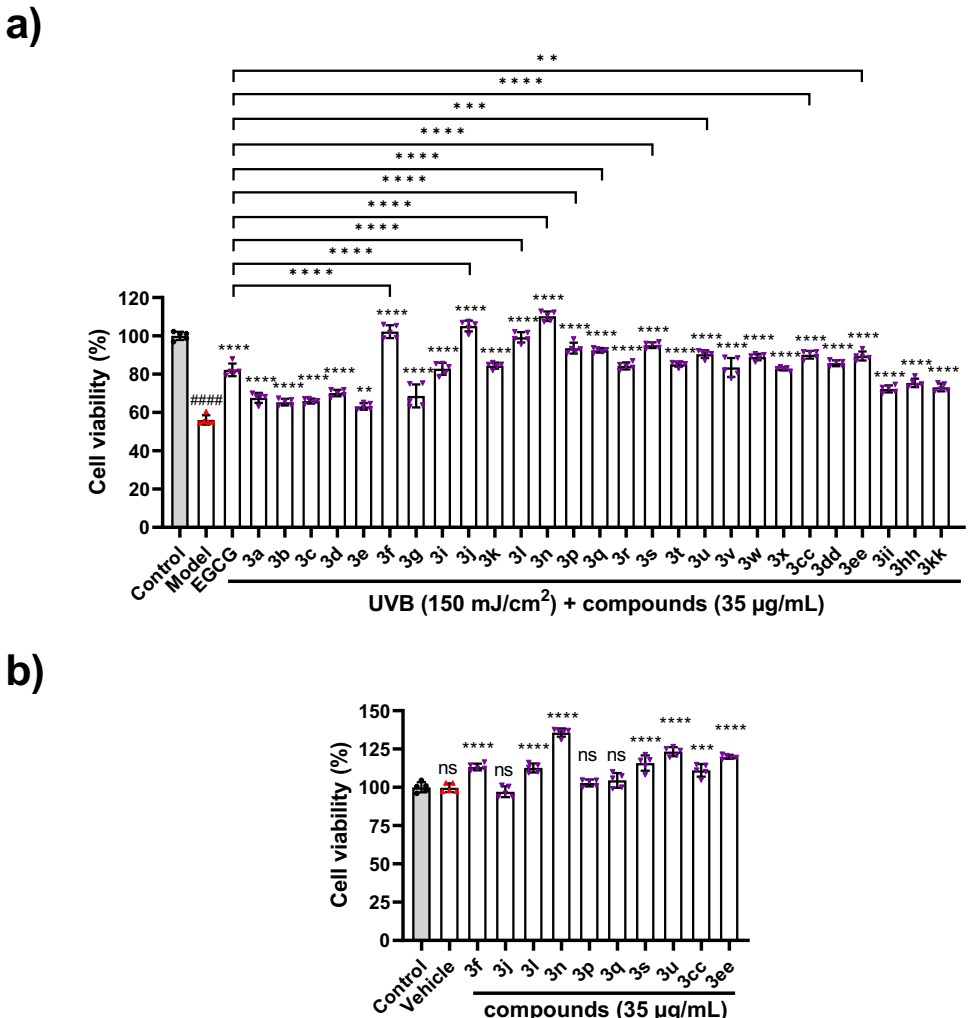

**Fig. 5 | *N*-Alkylated Indoles could protect HaCaT cells from UVB induced cell damage. a** HaCaT cells were firstly irradiated by 150 mJ/cm² UVB, then the cells were treated with 35 μg/mL different compounds or EGCG for 6 h and subsequently incubated with MTT for a further 4 h. Cell viability was measured by the absorbance of dissolved MTT crystals using an ELISA reader (*n* = 5). **b** The MTT assay was also used to measure the effects of the indicated compounds on the cell viability of HaCaT cells (*n* = 5). The data are representative of three independent experiments. All data were analyzed by software GraphPad Prism 9.5.0. Statistical significance between the indicated groups was calculated by using one-way ANOVA and multiple comparisons. Adjustments were made for multiple comparisons. Significance is presented as $^{####}p < 0.0001$ versus the control group and $^{**}p < 0.01$, $^{***}p < 0.001$, $^{****}p < 0.0001$ versus the model group or between the indicated groups in (**a**). Significance is presented as $^{***}p < 0.001$, $^{****}p < 0.0001$ versus the vehicle (DMSO) group in (**b**). ns: no significance ($p > 0.05$) versus the control or vehicle group. All the $p$ values adopted adjusted $p$ values.

and the structure of compound **8r** was characterized by X-ray. This unusual Markovnikov hydroalkylation reaction provided a method to indoles bearing a sterically demanding quaternary center, which are difficult to access (Fig. 6).

**Mechanistic studies**

The mechanism of CoH-catalyzed reductive hydroalkylation of 1,1-disubstituted alkenes remains elusive. Thus, a series of control experiments were performed to provide insight into the reaction mechanism (Fig. 7). First, the relationship between the enantiomeric excess (ee) of the ligand **L6** and the product **3a** was explored (Fig. 7a). A linear relationship was observed, suggesting that one ligand may coordinate with one cobalt atom to form the active catalyst. Next, a radical clock experiment was performed. When (iodomethyl)cyclopropane was subjected to standard conditions, the ring-opened product **9** was obtained whereas the cyclopropyl ring retained product **9'** was not obtained (Fig. 7b). These results indicate a radical pathway for activating alkyl halides. To further understand the reaction

mechanism, radical trapping experiments were conducted (Fig. 7c). The addition of TEMPO (2,2,6,6-tetramethyl-1-piperidinyloxy) significantly or totally inhibited the reaction (Fig. 7c), and the TEMPO adduct **10** was detected through HRMS-ESI. In addition to these, adduct **11** of benzyl radicals and TEMPO was detected under Condition B (Fig. 7c). This result led us to consider the possibility of the MHAT pathway, which was confirmed by subsequent DFT calculations. Deuterium-labeling experiments were conducted using Ph₂SiD₂ to investigate the regioselectivity (Fig. 7d). Deuterium was inserted in the α-position of **1a** or **1a'** in Condition A, whereas deuterium was inserted in the β-position of **1a'** in Condition B. These results indicate that the silane is the hydrogen source, and the ligand dominates the regioselectivity in the CoH-catalyzed reductive hydroalkylation. Based on the relevant references[44,45,47,49], we explored which of Co(I)H and Co(II)H is more probable in migratory insertion to alkenes, and the results of the UV-vis experiments showed slowly decreasing in the concentration of **L6**Co(II)X₂ (X = Br or I) with increasing yield, thus tending to favor the

**Fig. 6 | CoH-catalyzed Markovnikov hydroalkylation.** Condition B: [a]All reaction were carried out with **1'** (0.10 mmol), **2** (0.20 mmol, 2.0 equiv), CoI₂ (10 mol%), **L12** (12 mol%), CsF (0.20 mmol, 2.0 equiv), DEMS (0.20 mmol, 2.0 equiv), THF (0.5 mL, 0.2 M), 40 °C, 24 h. Isolated yields. The regioisomeric ratio (rr) > 20:1 in all cases. [b]Ph₂SiH₂ instead of DEMS. [c]CoBr₂·DME instead of CoI₂.

higher probability of alkene insertion by Co(II)H in the *anti*-Markovnikov hydroalkylation system, but it could not be completely exclude the possibility of Co(I)H insertion[47,49] (Fig. 7e).

## DFT calculations

Further DFT calculations were conducted to shed light on the mechanism of the reaction between Co-H complex and alkene, providing insights on the origin of enantioselectivity and regioselectivity. Given that the first-row transition metal complexes are known to have different spin states close in energies, both the quartet and doublet have been taken into accounts in our calculations[68]. We first examined the Co-H complex using **L6** as ligand. As shown in Fig. 8a, the migratory insertion transition states resulting in *anti*-Markovnikov selectivity were calculated to be much lower in energy, being in agreement with the experimental results. Conversely, in the Markovnikov migratory insertion transition state **²TS1_M-L6**, the methyl group of **1a** was observed to orient towards the aryl group of ligand, leading to substantial steric hindrance. Notably, **²TS1_S-aM-L6** was calculated to be more favored than **²TS1_R-aM-L6** by 2.7 kcal/mol which is consistent with the excellent enantioselectivity obtained in experiments. In **²TS1_S-aM-L6**, two substituents of alkene occupy less congested space with a π-π stacking stabilization identified between the indole group of **1a** and phenyl group of the ligand. On the contrary, in **²TS1_R-aM-L6**, steric repulsions between the methyl group and the ligand (H–H distance is 2.056 Å) leads an energy increase. In terms of spin state, our calculations showed that the migratory insertion transition states generally prefer to occur in doublet state rather than in quartet (see

Supplementary Fig. 20 for more details). Furthermore, considering the hydrogen atom transfer ability of Co-H complex, metal hydride H atom transfer (MHAT) processes have also been evaluated by DFT calculations. The quartet MHAT transition state **⁴TS2_M-L6** leading to Markovnikov product lies much higher than the doublet migratory insertion transition state in energy profile (see Supplementary Fig. 20 for more details). When **1a'** was utilized as the substrate in calculations (methyl group in **1a** is replaced by a phenyl group), similar trend was obtained. Specifically, the doublet *anti*-Markovnikov migratory insertion was predicted to be the more favorable than Markovnikov migratory insertion and MHAT processes (see Supplementary Fig. 21 for more details). Figure 8b shows the calculated energy profile using tridentate **L12** as ligand. Attempts to locate the migratory insertion transition state were unsuccessful as the congested isopropyl group hindered alkene coordination. On the other hand, the MHAT transition state is feasible due to a linear geometric arrangement. Intriguingly, our calculations show that although the doublet Co-H complex **²G** is more stable, the subsequent Markovnikov MHAT process occurs in the quartet via **⁴TS2_M-L12**, generating benzylic radical. This two-state reactivity scenario was commonly observed with the first-row transition metal complexes[68–72]. As shown in Fig. 8c, the calculated spin density for transition state of the Markovnikov MHAT suggests that during the reaction course, proton coupled with β electron was transferred towards alkene in doublet, while hydrogen atom possessing α electron was transferred in quartet. Compared with this Markovnikov MHAT-generating benzylic radical, the *anti*-Markovnikov MHAT in both spin states lies significantly higher on the energy profile due to the

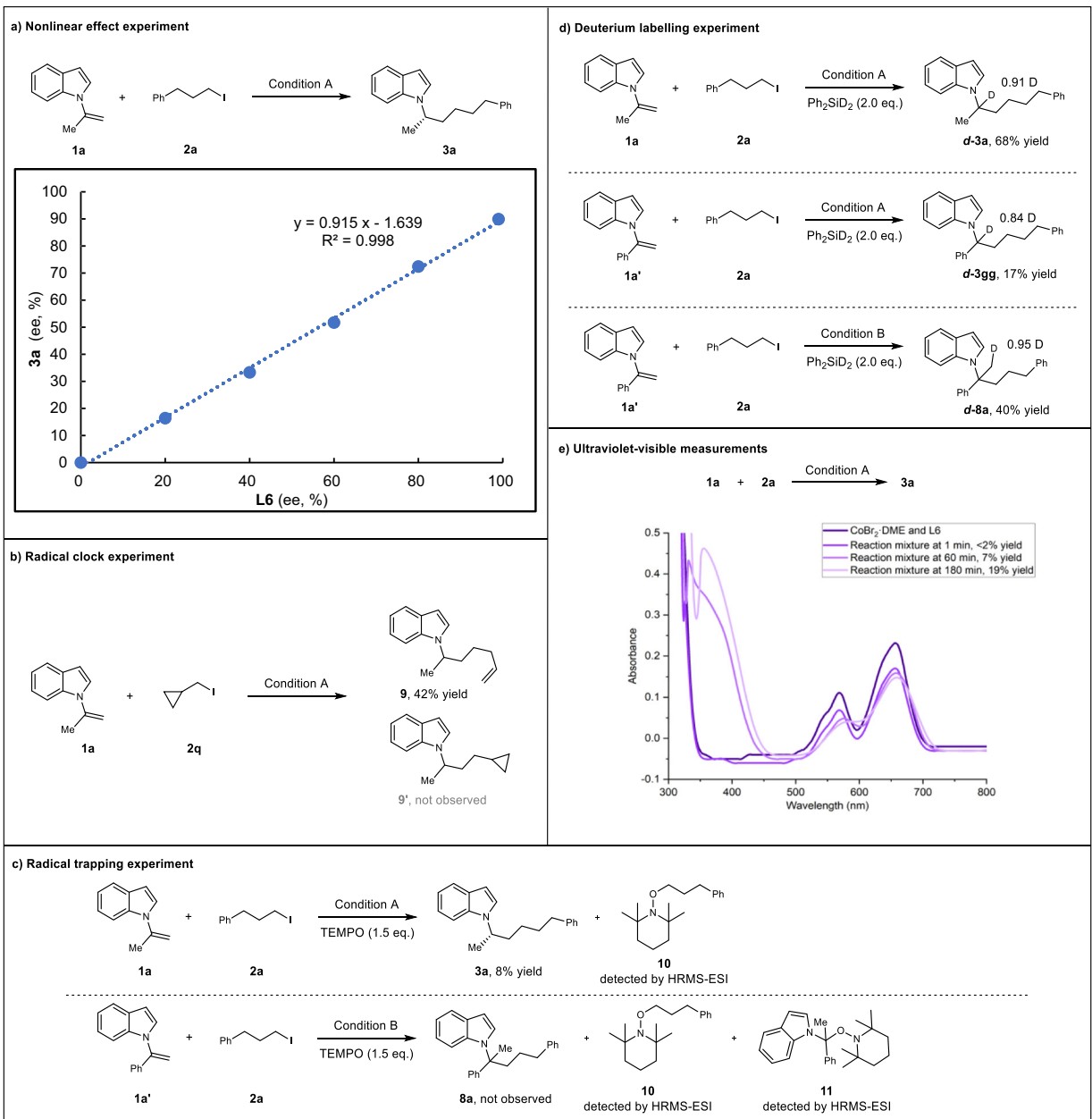

**Fig. 7 | Mechanistic studies. a** Nonlinear effect experiment. **b** Radical clock experiment. **c** Radical trapping experiment. **d** Deuterium labeling experiment. **e** Ultraviolet-visible measurements. TEMPO = 2,2,6,6-tetramethylpiperidinyloxy.

instability of the generated terminal radical. Following Markovnikov MHAT, the benzylic radical is trapped in the Co-alkyl complex **⁴H** via **⁴TS3_M-L12**. The combined radical trapping experiments and HRMS also confirmed the existence of the benzylic radical intermediate, supporting that the reaction undergoes through the MHAT process (see Supplementary Fig. 17 for more details). In summary, *anti*-Markovnikov migratory insertion was realized using the chiral BOX ligand **L6**, whose steric environments ensure both enantioselectivity and regioselectivity. When employing tridentate ligand **L12**, our calculations suggest that Markovnikov MHAT, liberating benzylic radical, is the most favorable pathway.

Based on previous literature[44–49,56,73], the above mechanistic investigations and DFT calculations, we propose a possible mechanism in Fig. 9. In the left-hand cycle, **L6**Co(I)X (**A**) which is considered as the initial active species, reaction with an alkyl halide to produce **L6**Co(II) X₂ (**B**) and the formation of an alkyl radical. Subsequently, **B** reacts with silane to form the metal hydride **L6**Co(II)XH (**C**), which inserts into the

C-C double bond to produce the primary *β*-alkylcobalt species (**D**), which captures the alkyl radical to obtain the Co(III) species **E**, then undergoes reductive elimination to produce the *anti*-Markovnikov hydroalkylation product and the initially active species **A**. In the right-hand cycle, **L12**Co(I) (**F**) is considered the initially active species and undergoes the same process as in the left-hand cycle to generate alkyl radicals and **L12**Co(II)X (**G**). **G** produces **L12**Co(II)H (**H**) in the presence of silanes, which subsequently undergoes the MHAT pathway, forming the cobalt species **I** with the release of benzyl radicals, and regenerates the alkyl cobalt species **J** (a process that is favored in DFT calculations), which then combines with the alkyl radicals to generate **K**, after reductive elimination produces the sterically crowded Markovnikov hydroalkylation product and the initially active species **F**.

## Discussion

In summary, we have achieved a general and enantioselective synthesis of a wide range of chiral *N*-heterocycles bearing a α,α-dialkyl

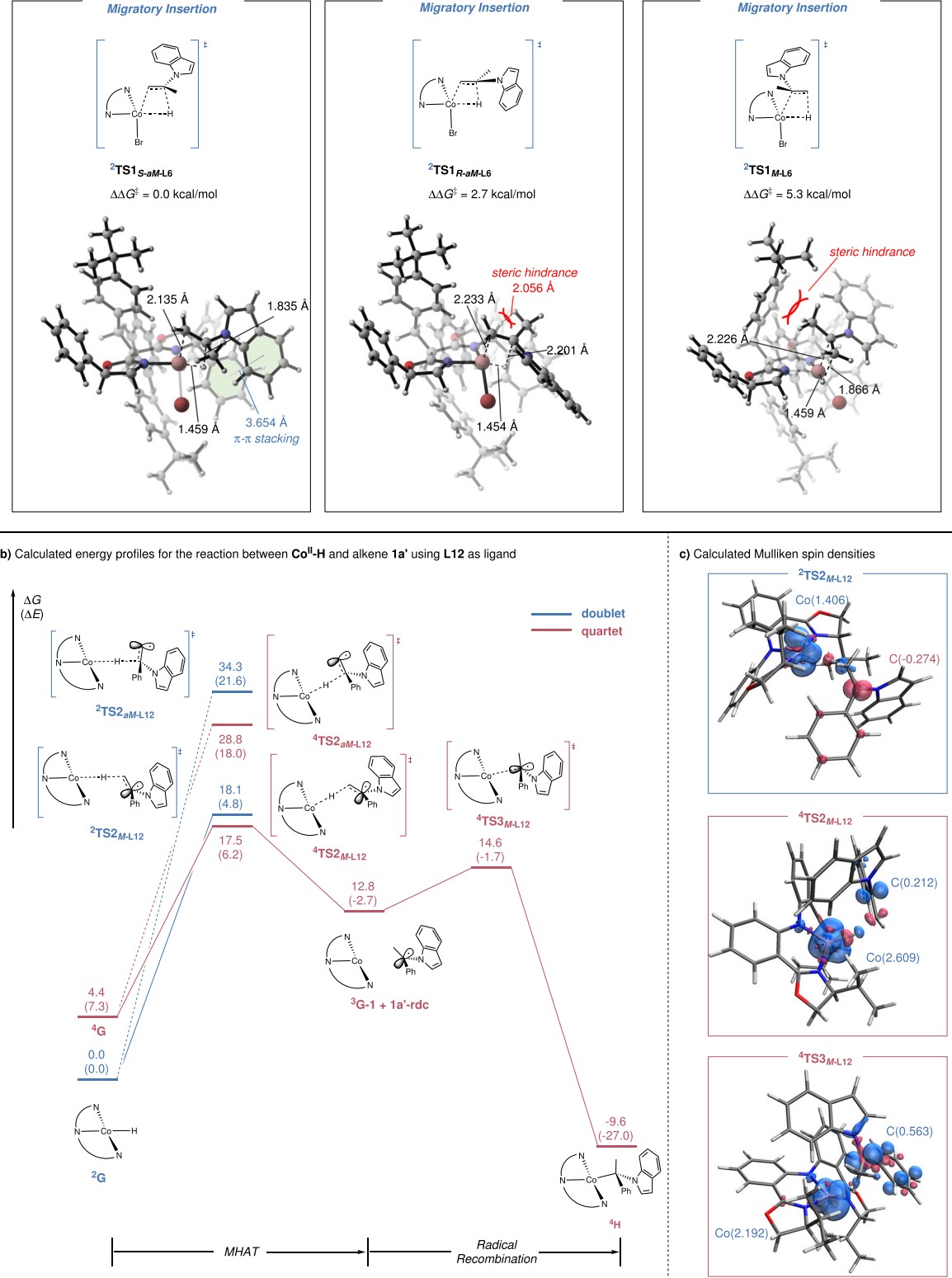

**Fig. 8 | DFT studies. a** Calculated enantiodetermining and regio-selective transition states using **L6** as ligand. **b** Energy profile on the MHAT process using **L12** as ligand; doublet is represented in blue line and quartet is depicted in red; the relative free energies and electronic energies (in parentheses) are given in kcal/mol. **c** Calculated Mulliken spin densities for transition states of MHAT and radical recombination; α electron densities are shown in blue lobe while β electron densities are plotted in red lobe.

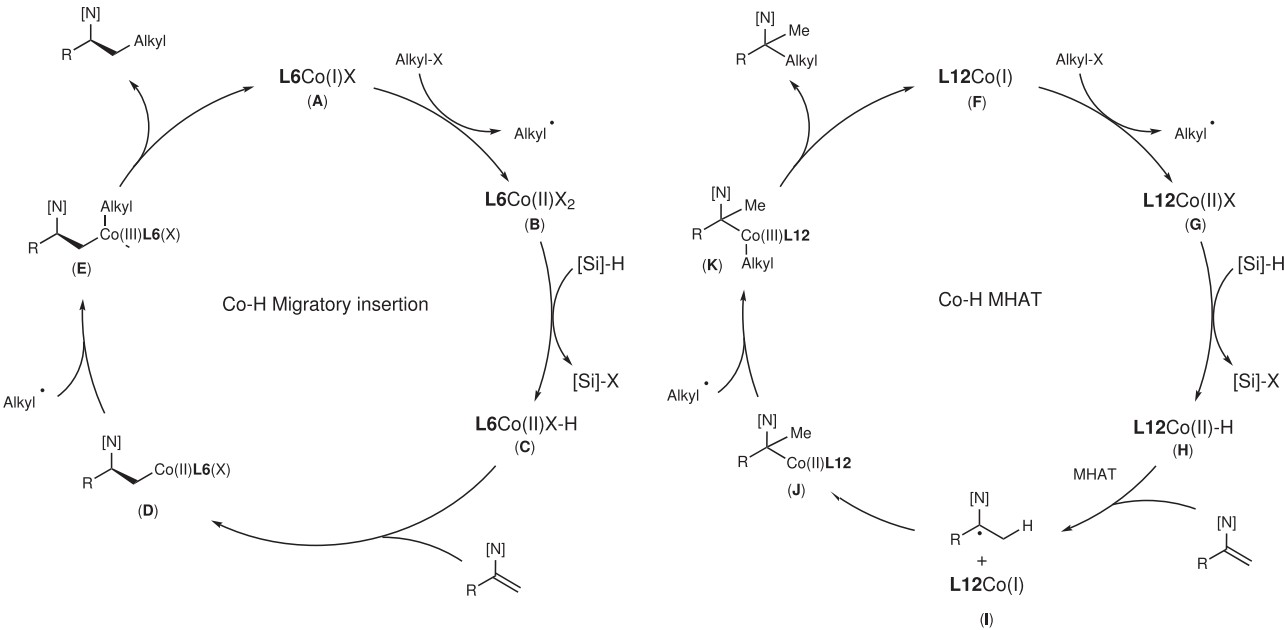

**Fig. 9 | Proposed reaction mechanism.** [N] = *N*-indolyl.

stereocenter from readily available and stable alkenes, by developing an unreported CoH-catalyzed asymmetric reductive hydroalkylation with high regio- and enantioselectivity. The key of this methodology is that the use of chiral Co-H catalysis, and the pyrrole group as a remote and weak directing group for stabilizing linear (primary) alkyl Co species intermediate to achieve distal stereocontrol. This work represents an rare example of metal hydride-catalyzed asymmetric hydroalkylation of 1,1-disubstituted alkenes via remote stereocontrol, and involves challenging linear chiral alkyl metal intermediates rather than conventional branched chiral alkyl metal intermediates. This asymmetric C(sp³)−C(sp³) coupling method features high flexibility in introducing a diverse set of alkyl groups at the *α*-position of chiral *N*-heterocycles and is able to produce chiral *N*-heterocycles with two minimally differentiated *α*-alkyl substituents. The utility of this methodology has been demonstrated by late-stage functionalization of drug molecules and the catalytic asymmetric synthesis of EZH2 inhibitor GSK126 and natural products (-)-Indolizidine 167B and (-)-Monomorine as well as conducting monomers. Notably, it also enable us to discover a class of molecules exhibiting anti-apoptosis activities in UVB-irradiated HaCaT cells. Low-valent Co-H catalysis in the alkene hydroalkylation has been under-exploited. However, the use of Co-H catalysis by changing the ligand from bi-dentate **L6** to tridentate **L12** enables CoH-catalyzed Markovnikov hydroalkylation, allowing to access indoles bearing a *α*-quaternary center. Combined mechanistic experiments and DFT calculations disclosed that the ligand played a crucial role in determining reaction mechanism which led to different regioselectivity. The *anti*-Markovnikov hydroalkylation involves a migratory insertion process while the Markovnikov hydroalkylation involves a MHAT process. This work represents cobalt-hydride catalyzed sterically-disfavored Markovnikov hydroalkylation of 1,1-disubstituted alkenes and also metal hydride-catalyzed regiodivergent hydroalkylation of 1,1-disubstituted alkenes. The present studies demonstrate the unique potential and advantages of Co-H catalysis in addressing challenging issues such as enantioselectivity and regioselectivity control of alkene hydroalkylation.

## Methods

### Representative procedure for the asymmetric hydroalkylation
Condition A: To an oven-dried 10.0 mL Teflon-screw cap test tube containing a magnetic stir was charged with CoBr₂•DME (3.1 mg, 10 mol%) and ligand **L6** (11.3 mg, 15 mol%) under a argon atmosphere using glove-box techniques. Subsequently, anhydrous DME (0.5 mL) was added, and the mixture was stirred for 10 minutes at room temperature. Then, CsF (30.4 mg, 0.20 mmol, 2.0 equiv.), **1** (0.10 mmol, 1.0 equiv), alkyl iodide **2** (0.20 mmol, 2.0 equiv.) and (OEt)₂MeSiH (32 uL, 0.20 mmol, 2.0 equiv.) were sequentially added. Afterwards, the tube was sealed with airtight electrical tapes and removed from the glove box and stirred at 0 °C for 24 hours at 500 rpm. After the reaction was completed, the reaction mixture was diluted with saturated NH₄Cl (aq., 2.0 mL) and EtOAc (3.0 mL). The aqueous phase was extracted with EtOAc (2 ×3.0 mL) and the combined organic phases were concentrated in vacuo. The crude mixture was purified by flash column chromatography on silica gel using a mixture of PE/EtOAc as eluent to obtain the desired product **3**.

### Representative procedure for the Markovnikov hydroalkylation
Condition B: An oven-dried 10.0 mL Teflon-screw cap test tube containing a magnetic stir was charged with CoI₂ (3.1 mg, 10 mol%) and ligand **L12** (4.7 mg, 12 mol%) under an argon atmosphere using glove-box techniques. Subsequently, anhydrous THF (0.5 mL) was added, and the mixture was stirred for 10 minutes at room temperature. Then, CsF (30.4 mg, 0.20 mmol, 2.0 equiv.), *N*-alkenyl indole **1'** (0.10 mmol, 1.0 equiv.), alkyl iodide **2** (0.20 mmol, 2.0 equiv.) and (OEt)₂MeSiH (32 uL, 0.20 mmol, 2.0 equiv.) were sequentially added. Afterwards, the tube was sealed with airtight electrical tapes and removed from the glove box and stirred at 40 °C for 24 hours at 500 rpm. After the reaction was completed, the reaction mixture was diluted with saturated NH₄Cl (aq., 2.0 mL) and EtOAc (3.0 mL). The aqueous phase was eacted with EtOAc (2 ×3.0 mL) and the combined organic phases were concentrated in vacuo. The crude mixture was purified by flash column chromatography on silica gel using a mixture of PE/EtOAc as eluent to obtain the desired product **8**.

### Reporting summary
Further information on research design is available in the Nature Portfolio Reporting Summary linked to this article.

## Data availability
The authors declare that the data supporting the findings of this study are available within the article and the Supplementary Information as

well as from the authors upon request. The coordinates of the optimized structure are available from the source data. The X-ray crystallographic coordinates for structure (**3o**) and (**8r**) reported in this study have been deposited at the Cambridge Crystallographic Data Centre (CCDC), under deposition numbers CCDC 2258190 and CCDC 2321724. These data can be obtained free of charge from The Cambridge Crystallographic Data Centre via www.ccdc.cam.ac.uk/data_request/cif. Source data are provided with this paper.

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

## Acknowledgements

This work was supported by the National Natural Science Foundation of China (22171240 and 22203034), Yunnan Natural Science Foundation (202301AU070206 and 202301AS070021), Yunling Scholars of Yunnan Province and the Postgraduate Research and Innovation Foundation of Yunnan University (No. KC-23234546). We thank Advanced Analysis and Measurement Center of Yunnan University for the sample testing service.

## Author contributions

Z.H.S. conceived and directed the project. J.R., Z.S., Y.W. and J.H.H. performed the chemistry experiments. S.Z., J.Y.H., C.Z. and R.Z. performed bioactivity experiments. Z.Z. performed the DFT studies. Z.H.S., Z.Z., C.H.Z. and X.J. analyzed the results and wrote the manuscript.

## Competing interests

The patent application (patent number: CN202311260532.4) was filed by Z.H.S., X.J., J.R., Z.S. and C.Z. from Yunnan University. The patent application covered through *anti*-Markovnikov hydroalkylation to access chiral α,α-dialkyl indoles and their application into effects on UVB-induced skin cell damage. The remaining authors declare no competing interests.
