## [Peer Review File · Nature Communications]

Enantioselective Synthesis of Chiral α,α -Dialkyl Indoles and Related Azoles by Cobalt-Catalyzed Hydroalkylation and Regioselectivity SwitchREVIEWER COMMENTS

Reviewer #1 (Remarks to the Author):

In this manuscript, Ren and co-workers reported Co(II)H- catalyzed hydroalkylation conditions to produce chiral N-heterocycles bearing an α -carbon stereocenter. Another highlight in this manuscript is that both anti-Markovnikov hydroalkylation and Markovnikov hydroalkylation products could be achieved by changing the ligands. It's an interesting phenomenon. However, no detailed mechanistic investigation or explanation was given in the manuscript. If this work could be done, I would recommend to publish the paper.

Additionally, some revisions should be done before formal publication:

1. Please carefully scan this manuscript. For example: (1) 2, Line 46: GSK126 (1); Line 48: Compound 2; Line 50: Compound 4. However, in Figure 1a, the compounds were listed by A, B, C, D. (2) Line 297, it should be 40 °C.
2. In Table 2, the Markovnikov hydroalkylation product 8 also contains a chiral carbon. However, no ee or er data was provided.
3. Line 128-131, "However, Co(I)H or Co(II)H catalyzed alkene hydroalkylation to form C(sp³)-C(sp³) bond has been under-exploited, and regiodivergent alkene hydroalkylation via low-valent Co(I)H or Co(II)H catalysis has not been achieved." In 2022, Yang and co-workers reported a Co(III)-H catalyzed anti-Markovnikov Hydroalkylation reaction using Co(acac)₃ and PhMeSiH₂. The authors should cite this paper and give a comparison.

Reviewer #2 (Remarks to the Author):

Shao described chiral α,α -dialkyl indoles and related azoles by cobalt-catalyzed hydroalkylation. The protocol enables the synthesis of highly valuable enantioenriched α,α -dialkyl N-heterocycles such as indoles, pyrroles, carbazoles, indolines and isatins. The optimization of reaction conditions is systematically demonstrated. The substrate scope is well studied. The most significant drawback of this manuscript is the mechanistic novelty. The fundamental reaction mechanism is known. Considering a point of this present work is asymmetric version, the author should discuss about the enantioselective model. Thus, the model supported by DFT calculations is necessary for clarifying the enantioselective mechanism. Overall, the novelty of this manuscript doesn't support publication as a communication in Nature Communications.

Reviewer #3 (Remarks to the Author):

Shao and Ji and co-workers reported a ligand-controlled cobalt-catalyzed regiodivergent hydroalkylation of 1,1-disubstituted vinylindole. The parts of anti-Markovnikov hydroalkylation have been presented as an asymmetric version, enabling the convenient synthesis of enantioenriched α,α -dialkyl indoles. anti-Markovnikov hydroalkylation of 1,1-disubstituted vinylindole has been applied to the diversification of drug molecules, catalytic asymmetric synthesis of EZH2 inhibitor GSK126, (-)-monomrine, (-)-indolizidine 167B and conducting monomers, demonstrating the high synthetic values. In addition, Markovnikov hydroalkylation was realized for the construction of quaternary carbon centers with a tridentate ligand, which was interesting. These reactions have good regio- and enantioselectivities and of course high coupling efficiencies, as well as excellent functional

group compatibility. Finally, the authors showed that α,α -dialkyl indoles could protect HaCaT cells from UVB-induced cell damage, expanding the readership of this manuscript.

The workload was extremely solid, and the experimental results were satisfactory. This reviewer strongly supports the publication of this manuscript in Nature Communications.

The following minor issues could be taken into consideration to further improve the scholarly presentation.

- 1) The comparison in Scheme 1 should be adjusted. The optimal conditions for nickel-hydride catalysis are often not the same as those for cobalt-hydride catalysis systems. In particular, DME is always the prevalent solvent for cobalt-hydride-catalyzed alkene hydroalkylation. However, both ether solvents (THF, Me-THF or some others) and amide solvents (DMAc or DMF) are suitable solvents for nickel-hydride-catalyzed alkene hydroalkylation. It is recommended to modify this section.
- 2) Shenvi reported a Mn/Ni dual catalysis to access the alkene hydroalkylation to form quaternary carbons (J. Am. Chem. Soc. 2019, 141, 7709). This literature precedent should be cited.
- 3) Secondary alkyl halides or activated alkyl halides (bromoacetates or benzyl chloride) could be examined. Even negative results would be informative for readers.
- 4) The authors showed that α,α -dialkyl indoles could protect HaCaT cells from UVB-induced cell damage. This reviewer was curious about the basis for choosing this test. Maybe brief and necessary pharmacological explanations and references could be added.
- 5) Cobalt has very rich valence states and interesting radical features. The mechanism for cobalt-catalyzed cross-coupling is always difficult to be clarified. Of course, it also includes the cobalt-catalyzed alkene hydroalkylation. In this submitted manuscript, the mechanism studies and rational citation of literature precedents are acceptable. Especially, the impressive regiodivergent synthesis and beautiful synthetic applications made this manuscript to be acceptable for publication in top journals, such as Nature Communications. However, this reviewer suggests some mechanism variants to be considered for discussion in the main text. For the mechanism of both anti-Markovnikov hydroalkylation and Markovnikov hydroalkylation, the pathway of Co(I)H insert into the alkenes to generate alkylcobalt species should be taken into consideration. This variant should be mentioned in the main text and related references could be cited. In addition, a hydrogen atom transfer to access tertiary benzyl radicals rather than the direct alkene hydrometallation should be taken into consideration for the Markovnikov hydroalkylation. This variant should be mentioned.
- 6) The format of the reference citation does not conform to Nature Communications. Nature Communications papers independently cite references one after another. In addition, there are a lot of examples of metal hydride-catalyzed enantioselective reductive hydroalkylation in Refs. 5. These references should be classified according to the metals (copper-hydride, nickel-hydride, cobalt-hydride, and iron-hydride) and sorted by the publication years. Moreover, the most related parts are the recent advances in cobalt-hydride-catalyzed enantioselective alkene hydroalkylation.
- 7) Several related references are suggested to be cited. A review on the recent advances in nickel-catalyzed reductive hydroalkylation and hydroarylation of electronically unbiased alkenes, *Sci. China Chem.* 2020, 63, 1586. A related example on Ligand-controlled cobalt-catalyzed regiodivergent alkyne hydroalkylation, *J. Am. Chem. Soc.* 2022, 144, 13961. A recent progress on iron-catalyzed alkene hydroalkylation, *Angew. Chem. Int. Ed.* 2023, 62, e202306663.
- 8) There are some clerical errors in the main text and references of the article, which need to be checked and corrected. a) In lines 46-51, the number for the example molecular should be A, B, C, D as shown in Figure 1, instead of 1, 2, 3, 4. b) There is an error in reference 5k, which appears to be two references.

Thank three reviewers very much for the comments, suggestions and nice consideration of our manuscript for publication in Nature Communications Journal. We have made the revisions in a genuine effort to address the concerns. The point-by-point responses are listed below. Please see the changes in the revised manuscript text file with yellow colour highlighting.

Before listing point-by-point responses to reviewers, we would like to point that according to the journal style requirement, the manuscript title “Enantioselective Synthesis of Chiral α,α -Dialkyl Indoles and Related Azoles by Cobalt-Catalyzed Hydroalkylation: Reaction Development, Regioselectivity Switch and Applications” has been adjusted into “Synthesis of Chiral α,α -Dialkyl Indoles and Related Azoles by Cobalt-Catalyzed Hydroalkylation and Regioselectivity Switch”. In addition, Abstract has been shortened to less 150 words.

Reviewer 1:

In this manuscript, Ren and co-workers reported Co(II)H-catalyzed hydroalkylation conditions to produce chiral N-heterocycles bearing an α -carbon stereocenter. Another highlight in this manuscript is that both anti-Markovnikov hydroalkylation and Markovnikov hydroalkylation products could be achieved by changing the ligands. It’s an interesting phenomenon. However, no detailed mechanistic investigation or explanation was given in the manuscript. If this work could be done, I would recommend to publish the paper.

Our response: We thank sincerely this reviewer for very instructive comments and suggestions. According to the comments and suggestions of this reviewer, we have made the corresponding investigations. Notably, these investigations disclosed interesting mechanistic information. The detailed mechanistic investigation and explanation have been given in the revised manuscript.

In our previous mechanism studies, we confirmed the existence of alkyl radical intermediates in both systems using the radical clock experiment, the radical trapping experiment and the deuterium labelling experiment. In addition, different ligands played a crucial role in the interaction of Co-H species with alkenes. We further investigated the system of Markovnikov hydroalkylation, we detected benzyl radical combined with TEMPO compound **11** by HRMS (see Figure 5c), a result that shows the possibility of the existence of the MHAT mechanism. In the *anti*-Markovnikov hydroalkylation system, based on the relevant references (*Nat. Catal.* **2021**, *4*, 901-911; *J. Am. Chem. Soc.* **2022**, *144*, 13961-13972; *Angew. Chem. Int. Ed.* **2023**, *62*, e202306381; *J. Am. Chem. Soc.* **2024**, *146*, 3405-3415), we performed UV-visible experiments, and the results indicated a higher probability of Co(II)H species during the insertion of Co-H species into alkenes (see Figure 5e). To further shed light on the mechanism of the reaction between Co-H complex and alkene, we have carried out DFT calculations, providing insights on the origin of enantioselectivity and regioselectivity and explaining why the product is obtained as an *anti*-Markovnikov addition when **L6** is used while the product is obtained as a Markovnikov addition when **L12** is used. The detailed computational results have been added to the manuscript as well as Supplementary Information (see Figure 5 and Figure 6 in the manuscript, other information is provided on pages S86-S87 of the Supplementary Information). In summary, the *anti*-Markovnikov hydroalkylation involving a migratory insertion process was realized using the chiral BOX ligand **L6** whose steric environments

ensure both enantioselectivity and regioselectivity. When employing tridentate ligand **L12**, our calculations suggest that the Markovnikov hydroalkylation involving a MHAT process is the most favorable pathway (see Figure 6 and Figure 7 in the manuscript). Based on mechanistic experiments and DFT calculations, we reconsidered the reaction mechanism and the Co-H catalyzed MHAT process is more acceptable when using **L12**.

Additionally, some revisions should be done before formal publication:

1. Please carefully scan this manuscript. For example: (1) 2, Line 46: GSK126 (1); Line 48: Compound 2; Line 50: Compound 4. However, in Figure 1a, the compounds were listed by A, B, C, D. (2) Line 297, it should be 40 °C.

Our response: Many thanks. We have carefully scanned this manuscript and made the corresponding corrections. Regarding line 297, we re-checked it, and it should be 0 °C. The temperature of Markovnikov hydroalkylation is 40 °C. Thanks again.

2. In Table 2, the Markovnikov hydroalkylation product **8** also contains a chiral carbon. However, no ee or er data was provided.

Our response: Thank this reviewer very much. The Markovnikov hydroalkylation product **8a** has an er value of only 65:35 as determined by chiral HPLC. In view of the poor enantioselectivity, we have not listed the corresponding er values in Table 2, and the results relating to the product **8a** have been shown on Figure S6 of SI.

3. Line 128-131, “However, Co(I)H or Co(II)H catalyzed alkene hydroalkylation to form C(sp³)-C(sp³) bond has been under-exploited, and regiodivergent alkene hydroalkylation via low-valent Co(I)H or Co(II)H catalysis has not been achieved.” In 2022, Yang and co-workers reported a Co(III)-H catalyzed anti-Markovnikov Hydroalkylation reaction using Co(acac)₃ and PhMeSiH₂. The authors should cite this paper and give a comparison.

Our response: Thanks. According to the suggestion of this reviewer, we have cited this paper and added the sentence “A high-valent Co(IV)H-catalyzed *anti*-Markovnikov non-enantioselective hydroalkylation of alkenes with oxime esters was recently reported by introducing an 8-aminoquinoline as the directing group⁴³. In this directed process, alkyl Co intermediates were not be involved.”

Reviewer 2:

Shao described chiral α,α -dialkyl indoles and related azoles by cobalt-catalyzed hydroalkylation. The protocol enables the synthesis of highly valuable enantioenriched α,α -dialkyl N-heterocycles such as indoles, pyrroles, carbazoles, indolines and isatins. The optimization of reaction conditions is systematically demonstrated. The substrate scope is well studied. The most significant drawback of this manuscript is the mechanistic novelty. The fundamental reaction mechanism is known. Considering a point of this present work is asymmetric version, the author should discuss about the enantioselective model. Thus, the model supported by DFT calculations is necessary for clarifying the enantioselective mechanism. Overall, the novelty of this manuscript doesn't support publication as a communication in Nature Communications.

Our response: We sincerely appreciate this reviewer's invaluable comments and suggestions, promoting us to conduct more mechanistic studies (especially DFT calculations) that led to new and interesting discovery. Mechanistic experiments and DFT calculations disclosed that the ligand played a crucial role in dictating reaction mechanism which led to different regioselectivity outcomes by changing the interaction modes of Co-H species with alkenes. We found that by changing the type of ligand, the reaction mechanism is changed. The Co(II)H-catalyzed *anti*-Markovnikov hydroalkylation of 1,1-disubstituted alkenes **IV** with bi-dentate ligand **L6** involves a migratory insertion process whereas interestingly the Co(II)H-catalyzed Markovnikov hydroalkylation of 1,1-disubstituted alkenes **IV** with tridentate ligand **L12** involves a metal-hydride hydrogen atom transfer (MHAT) process rather than a migratory insertion process that we proposed in previous submission. Such an interesting chemistry has not yet been discovered in metal-hydride catalyzed hydroalkylation of alkenes. It is noted that in previous literatures of low-valent Co(I)H or Co(II)H-catalyzed hydroalkylation reactions of alkenes, the reactions involved the migratory insertion process. The hydroalkylation of alkenes with alkyl halide via the MHAT strategy catalyzed by Co-H has yet not been reported previously. Here, we would also like to mention that besides novel mechanism, sterically-disfavoured Markovnikov hydroalkylation reaction of 1,1-disubstituted alkenes catalyzed by the cobalt-hydride that is achieved in this paper is an important advance in this field. Because of steric hindrance, facile β -hydrogen elimination, and ease of isomerization of tertiary alkylmetals, metal hydride-catalyzed sterically-disfavoured Markovnikov hydroalkylation of 1,1-disubstituted alkenes remains an elusive challenge. This work is the first cobalt-hydride catalyzed sterically-disfavoured Markovnikov hydroalkylation of 1,1-disubstituted alkenes. In addition, it is also the first metal hydride-catalyzed regiodivergent hydroalkylation of 1,1-disubstituted alkenes, and also the first metal hydride-catalyzed regiodivergent hydroalkylation of alkenes, in which ligands changed the reaction mechanism which led to regiodivergence. Reaction strategies allowing regiodivergence represents one of the most cutting-edge developments in synthetic organic chemistry and medicinal chemistry.

On the other hand, DFT calculations also provided insights on the origin of enantioselectivity and established the enantioselective model for understanding the enantiocontrol in the Co-H catalyzed asymmetric hydroalkylation of 1,1-disubstituted alkenes, which remains elusive. Metal hydride (MH)-catalyzed enantioselective hydroalkylation of alkenes with alkyl electrophiles has emerged as an attractive strategy toward value-added chemicals and medicinally relevant compounds, which allows for simultaneous construction of alkyl-alkyl ($C(sp^3)-C(sp^3)$) bond and generation of a new stereocenter. However, 1,1-disubstituted alkenes have rarely been successfully employed in metal-hydride-catalyzed enantioselective hydroalkylation reactions. Unlike mono-substituted and 1,2-disubstituted alkenes involving a branched (secondary) alkyl metal intermediates of the type **II**, M-H catalyzed asymmetric hydroalkylation of 1,1-disubstituted alkenes would involve linear alkyl Co intermediates of the type **III**, and would require remote stereocontrol. The M-H catalyzed asymmetric hydroalkylation of 1,1-disubstituted alkenes remains an elusive challenge. The key challenge is the need to identify a catalytic system capable of not only efficient and regioselective M-H insertion of these sterically encumbered substrates but also effective discrimination of olefin substituents distant from the metal center and its chiral ligand. Thus, the first CoH-catalyzed highly enantioselective hydroalkylation reaction of 1,1-disubstituted alkenes that is achieved in this paper constitutes another important advance in this field.

Overall, we believe that the revision significantly improved the quality of our manuscript, and meet the standard as an article in Nature Communications Journal. Thanks again for your invaluable comments and suggestions, and nice consideration of this manuscript for publication in NC.

Reviewer 3:

Shao and Ji and co-workers reported a ligand-controlled cobalt-catalyzed regio-divergent hydroalkylation of 1,1-disubstituted vinylindole. The parts of anti-Markovnikov hydroalkylation have been presented as an asymmetric version, enabling the convenient synthesis of enantioenriched α,α -dialkyl indoles. anti-Markovnikov hydroalkylation of 1,1-disubstituted vinylindole has been applied to the diversification of drug molecules, catalytic asymmetric synthesis of EZH2 inhibitor GSK126, (-)-monomorphine, (-)-indolizidine 167B and conducting monomers, demonstrating the high synthetic values. In addition, Markovnikov hydroalkylation was realized for the construction of quaternary carbon centers with a tridentate ligand, which was interesting. These reactions have good regio- and enantioselectivities and of course high coupling efficiencies, as well as excellent functional group compatibility. Finally, the authors showed that α,α -dialkyl indoles could protect HaCaT cells from UVB-induced cell damage, expanding the readership of this manuscript.

The workload was extremely solid, and the experimental results were satisfactory. This reviewer strongly supports the publication of this manuscript in Nature Communications.

The following minor issues could be taken into consideration to further improve the scholarly presentation.

1. The comparison in Scheme 1 should be adjusted. The optimal conditions for nickel-hydride catalysis are often not the same as those for cobalt-hydride catalysis systems. In particular, DME is always the prevalent solvent for cobalt-hydride-catalyzed alkene hydroalkylation. However, both ether solvents (THF, Me-THF or some others) and amide solvents (DMAc or DMF) are suitable solvents for nickel-hydride-catalyzed alkene hydroalkylation. It is recommended to modify this section.

Our response: Thank this reviewer for the suggestion. We have modified this section. Accordingly, Scheme 1 was replaced with Table 1. The sentences “In initial studies, we applied the nickel catalysts for the hydroalkylation reaction of **1a** and **2a** (Scheme 1), given that NiH catalysis has found wide application in the alkene hydroalkylation. Unfortunately, we obtained only a trace of the product **3a**. We therefore turned our attention to the use of other metal salts, and we found that $\text{CoBr}_2 \cdot \text{DME}$ as the metal salt in combination with the chiral ligand **L6** offered the *anti*-Markovnikov hydroalkylation product **3a** in 98% yield with 95:5 er (see the Supporting Information for the details).” have been adjusted into “In initial studies, we applied nickel catalysts for the hydroalkylation reaction of **1a** and **2a**, given that Ni-H catalysis has found wide application in the alkene hydroalkylation. Unfortunately, we obtained only 8% yield and 69:31 er of the *anti*-Markovnikov hydroalkylation product **3a** with DME as the solvent (Table 1, entry 1). Only a trace of product **3a** was detected by using THF, DMA and DMF, which are frequently used solvents in Ni-H catalysis (entries 2-4). We therefore turned our attention to the use of other metal salts, and found that $\text{CoBr}_2 \cdot \text{DME}$ as the metal salt and DME as the solvent in combination with the chiral

ligand **L6** offered the *anti*-Markovnikov hydroalkylation product **3a** in 98% yield with 95:5 er (entry 5) (see the Supplementary Information Table S1 for the details)."

We reran the Ni-H vs. Co-H catalyzed hydroalkylation comparison in Table 1, and the results show that in this reaction system, despite the use of solvents commonly used for Ni-H catalysis, the Co-H catalyzed results are still significantly better than the Ni-H catalyzed results.

2. Shenvi reported a Mn/Ni dual catalysis to access the alkene hydroalkylation to form quaternary carbons (J. Am. Chem. Soc. 2019, 141, 7709). This literature precedent should be cited.

Our response: According to the suggestion of this reviewer, this literature has been cited. Please see the ref. 56.

3. Secondary alkyl halides or activated alkyl halides (bromoacetates or benzyl chloride) could be examined. Even negative results would be informative for readers.

Our response: According to the suggestion of this reviewer, we have examined the *anti*-Markovnikov hydroalkylation reactions with secondary alkyl halides and activated alkyl halides (bromoacetates and benzyl chloride). Please see the Supplementary Information Figure S5. We added the sentence "Iodocyclohexane was found to be less reactive (see the Supplementary Information Figure S5)" in the main text of the manuscript.

Meanwhile, we have also examined the Markovnikov hydroalkylation reactions with secondary alkyl halides and activated alkyl halides (bromoacetates and benzyl chloride). Interestingly, benzyl chlorides were able to provide the corresponding Markovnikov hydroalkylation products in acceptable yields. Iodocyclohexane and benzyl 2-bromoacetate were found to be not reactive. We added the sentences "Interestingly, benzyl chlorides were able to provide the corresponding Markovnikov hydroalkylation products **8r-8u** in acceptable yields (see the Supplementary Information Figure S6), and the structure of compound **8r** was characterized by X-ray." in the main text of the manuscript.

4. The authors showed that α,α -dialkyl indoles could protect HaCaT cells from UVB-induced cell damage. This reviewer was curious about the basis for choosing this test. Maybe brief and necessary pharmacological explanations and references could be added.

Our response: Thanks for the reviewer's suggestion. In this study, we synthesized a series of new *N*-alkylated indoles, and we hoped to find new applications of these molecules. Frankly, it remains unknown whether they have photoprotective effects on UVB-induced skin cell damage. Our co-workers in School of Pharmacy of Yunnan University has a good experience in screening the active components for protecting the skin cells from UV triggered damage, so we performed experiments for detecting the protective effects of *N*-alkylated indoles on UV induced skin injury. Pleasingly, we discovered a new class of molecules exhibiting *anti*-apoptosis activities in UVB-irradiated HaCaT cells.

Indole is a heterocycle commonly found in natural products and has many biological activities, such as antibacterial^[1], antifungal^[2], antiinflammatory^[3], antihistamine^[4], antioxidant^[5], antidiabetic^[6], antiviral^[7] and anticancer activities, etc^[8-10]. Skin is an important barrier to protect the body from environmental stress. However, exposure to ultraviolet radiation (UV) and various environmental oxidative stresses can cause skin inflammation^[11,12]. Indole derivatives, such as melatonin and

indole-3-lactic acid (ILA), could reduce the cytotoxicity caused by UVB irradiation and suppress the production of pro-inflammatory cytokines in UVB-irradiated keratinocytes^[13-15]. To further demonstrate the utility of new *N*-alkylated indoles that we synthesized in this study, we performed experiments for detecting the protective effects of *N*-alkylated indoles on UV induced skin injury.

According to the reviewer's suggestion, we have added the sentences "Skin is an important barrier to protect the body from environmental stress. However, exposure to ultraviolet radiation (UV) and various environmental oxidative stresses can cause skin inflammation^{63,64}. Indole derivatives, such as melatonin and indole-3-lactic acid (ILA), could reduce the cytotoxicity caused by UVB irradiation and suppress the production of pro-inflammatory cytokines in UVB-irradiated keratinocytes⁶⁵⁻⁶⁷. In this study, we synthesized a series of new *N*-alkylated indoles, but it remained unknown whether they have photoprotective effects on UVB-induced skin cell damage."

References:

- [1] Konus, M. et al. Synthesis, biological evaluation and molecular docking of novel thiophene-based indole derivatives as potential antibacterial, GST inhibitor and apoptotic anticancer agents. *ChemistrySelect*. **5**, 5809–5814 (2020).
- [2] Sivaprasad, G., Perumal, P. T., Prabavathy, V. R. & Mathivanan, N. Synthesis and antimicrobial activity of pyrazolylbisindoles—Promising anti-fungal compounds. *Bioorg. Med. Chem. Lett.* **16**, 6302–6305 (2006).
- [3] Sondhi, S. M., Dinodia, M. & Kumar, A. Synthesis, anti-inflammatory and analgesic activity evaluation of some amidine and hydrazone derivatives. *Bioorg. Med. Chem.* **14**, 4657–4663 (2006).
- [4] Battaglia, S. et al. Indole amide derivatives: synthesis, structure–activity relationships and molecular modelling studies of a new series of histamine H₁-receptor antagonists. *Eur. J. Med. Chem.* **34**, 93–105 (1999).
- [5] Sharma, V., Kumar, P. & Pathak, D. Biological importance of the indole nucleus in recent years: A comprehensive review. *J. Heterocycl. Chem.* **47**, 491–502 (2010).
- [6] Zhu, Y. et al. Research progress of indole compounds with potential antidiabetic activity. *Eur. J. Med. Chem.* **223**, 113665 (2021).
- [7] Abdel-Gawad, H., Mohamed, H. A., Dawood, K. M. & Badria, F. A.-R. Synthesis and Antiviral Activity of New Indole-Based Heterocycles. *Chem. Pharm. Bull.* **58**, 1529–1531 (2010).
- [8] Vicini, P. et al. Synthesis and antiproliferative activity of benzo[*d*]isothiazole hydrazones. *Eur. J. Med. Chem.* **41**, 624–632 (2006).
- [9] Terzioglu, N. & Gürsoy, A. Synthesis and anticancer evaluation of some new hydrazone derivatives of 2,6-dimethylimidazo[2,1-*b*][1,3,4]thiadiazole-5-carbohydrazide. *Eur. J. Med. Chem.* **38**, 781–786 (2003).
- [10] Abadi, A. H., Eissa, A. A. H. & Hassan, G. S. Synthesis of novel 1,3,4-trisubstituted pyrazole derivatives and their evaluation as antitumor and antiangiogenic agents. *Chem. Pharm. Bull.* **51**, 838–844 (2003).
- [11] Rittié, L. & Fisher, G. J. UV-light-induced signal cascades and skin aging. *Ageing Res. Rev.* **1**, 705–720 (2002).
- [12] Rinnerthaler, M., Bischof, J., Streubel, M. K. Trost, A. & Richter, K. Oxidative Stress in Aging Human Skin. *Biomolecules.* **5**, 545–589 (2015).

- [13] Nickel, A. & Wohlrab, W. Melatonin protects human keratinocytes from UVB irradiation by light absorption. *Arch. Dermatol. Res.* **292**, 366–368 (2000).
- [14] Fischer, T. W., Slominski, A., Zmijewski, M. A., Reiter, R. J. & Paus, R. Melatonin as a major skin protectant: from free radical scavenging to DNA damage repair. *Exp. Dermatol.* **17**, 713–730 (2008).
- [15] Aoki-Yoshida, A. et al. Prevention of UVB-induced production of the inflammatory Mediator in human keratinocytes by lactic acid derivatives generated from aromatic amino acids. *Biosci., Biotechnol., Biochem.* **77**, 1766–1768 (2013).

5. Cobalt has very rich valence states and interesting radical features. The mechanism for cobalt-catalyzed cross-coupling is always difficult to be clarified. Of course, it also includes the cobalt-catalyzed alkene hydroalkylation. In this submitted manuscript, the mechanism studies and rational cation of literature precedents are acceptable. Especially, the impressive regiodivergent synthesis and beautiful synthetic applications made this manuscript to be considered for discussion in the main text. For the mechanism of both anti-Markovnikov hydroalkylation and Markovnikov hydroalkylation, the pathway of Co(I)H insert into the alkenes to generate alkylcobalt species should be taken into consideration. This variant should be mentioned in the main text and related references could be cited. In addition, a hydrogen atom transfer to access tertiary benzyl radicals rather than the direct alkene hydrometallation should be taken into consideration for the Markovnikov hydroalkylation. This variant should be mentioned.

Our response: Thank very much this reviewer for instructive suggestions. In the system of Markovnikov hydroalkylation, we detected benzyl radical combined with TEMPO compound **11** by HRMS (see the Figure 5c), a result that shows the possibility of the existence of the MHAT mechanism. In the *anti*-Markovnikov hydroalkylation system, based on the relevant references (*Nat. Catal.* **2021**, *4*, 901-911; *J. Am. Chem. Soc.* **2022**, *144*, 13961-13972; *Angew. Chem. Int. Ed.* **2023**, *62*, e202306381; *J. Am. Chem. Soc.* **2024**, *146*, 3405-3415), we performed UV-visible experiments, and the results indicated a higher probability of Co(II)H species during the insertion of Co-H species into alkenes (see the Figure 5e). To further shed light on the mechanism of the reaction between Co-H complex and alkene, we have carried out DFT calculations, providing insights on the origin of enantioselectivity and regioselectivity. The detailed computational results have been added to the manuscript as well as Supplementary Information. Given that the first-row transition metal complexes are known to have different spin states close in energies, both the quartet and doublet have been taken into accounts in our calculations. Interestingly, we found that the ligand played a crucial role in determining reaction mechanism which led to different regioselectivity. In summary, the *anti*-Markovnikov hydroalkylation involving the migratory insertion process was realized using the chiral BOX ligand **L6** whose steric environments ensure both enantioselectivity and regioselectivity. When employing tridentate ligand **L12**, our calculations suggest that the Markovnikov MHAT as suggested by this reviewer, liberating benzylic radical, is the most favorable pathway. The combined radical trapping experiments and HRMS also confirmed the existence of the benzylic radical intermediate, supporting that the reaction undergoes through MHAT process. The other variant of the reaction mechanism mentioned by this reviewer has been updated in the manuscript.

6. The format of the reference citation does not conform to Nature Communications. Nature Communications papers independently cite references on after another. In addition, there are a lot

of examples of metal hydride-catalyzed enantioselective reductive hydroalkylation in Refs.5. These references should be classified according to the metals (copper-hydride, nickel-hydride, cobalt-hydride, and iron-hydride) and sorted by the publication years. Moreover, the most related parts are the advances in cobalt-hydride-catalyzed enantioselective alkene hydroalkylation.

Our response: Thanks this reviewer for the comments and suggestions, we have corrected the reference style and categorized the relevant literature according to the type of metal and the year of publication. Accordingly, in the third paragraph of Introduction section, we have added the sentence “Depending on the type of metal, it can be categorized as copper-hydride²³, nickel-hydride²⁴⁻⁴², cobalt-hydride⁴³⁻⁴⁹, and iron-hydride⁵⁰.”

7. Several related references are suggested to be cited. A review on the recent advances in nickel-catalyzed reductive hydroalkylation and hydroarylation of electronically unbiased alkenes, *Sci. China Chem.* **2020**, 63, 1586. A related example on Ligand-controlled cobalt-catalyzed regiodivergent alkyne hydroalkylation, *J. Am. Chem. Soc.* **2022**, 144, 13961. A recent progress on iron-catalyzed alkene hydroalkylation, *Angew. Chem. Int. Ed.* **2023**, 62, e202306663.

Our response: Thanks this reviewer for the suggestion, we have cited these references. Please see refs. 18, 45 and 50.

8. There are some clerical errors in the main text and references of the article, which need to be checked and corrected. a) In lines 46-51, the number for the example molecular should be A, B, C, D as shown in Figure 1, instead of 1, 2, 3, 4. b) There is an error in reference 5k, which appears to be two references.

Our response: Thanks. We have checked and corrected these errors.

REVIEWERS' COMMENTS

Reviewer #1 (Remarks to the Author):

In the revised manuscript, Ren and co-workers have made the revisions according to the three reviewers' comments. The work is solid and meets the standard for publication in Nature Communication.

Reviewer #2 (Remarks to the Author):

The revised manuscript is suitable for publication.

Reviewer #3 (Remarks to the Author):

I carefully read the author's response letter and the revised manuscript. I believe that the author has made significant modifications to the manuscript based on the suggestions and comments of the three reviewers, resulting in a noticeable improvement in quality. I strongly recommend that the revised manuscript be published in Nature Communications.